# COVID-19-Related Stress, Fear and Online Teaching Satisfaction among Nursing Students during the COVID-19 Pandemic

**DOI:** 10.3390/healthcare11060894

**Published:** 2023-03-20

**Authors:** Sanja D. Tomić, Slobodan Tomić, Goran Malenković, Jelena Malenković, Armin Šljivo, Ermina Mujičić

**Affiliations:** 1Faculty of Medicine, University of Novi Sad, 21000 Novi Sad, Serbia; 2Clinical Center of University of Sarajevo, 71000 Sarajevo, Bosnia and Herzegovina

**Keywords:** COVID-19, mental health, stress, cognition, quality of life

## Abstract

The COVID-19 pandemic has had a significant impact on mental health, particularly among students, due to COVID-19-related fear and also the transition from traditional to online lectures. In this questionnaire-based study, the COVID-19 Stress Scales (CSS), the Fear of COVID-19 Scale (FCV-19S), and the Online Teaching Satisfaction Scale were used to assess COVID-19-related fear, stress, and overall satisfaction with online teaching during the COVID-19 pandemic among nursing students in Serbia. A total of 167 students participated in the study, whose mean age was 21.3 ± 5.3, and the majority of whom were female and first-year students. Overall, most students experienced moderate to extremely high COVID-19-related stress levels. Overall, first-year and fourth-year students scored significantly lower regarding the Xenophobia and Traumatic stress subscales than second-year and third-year students, whereas first-year students also scored significantly lower on the Danger and Contamination subscales. First-year students experienced less COVID-19-related fear compared to senior students. Students were reasonably satisfied with online teaching. A stratified program is needed to prevent further decline of students’ mental health and to improve their adaptation through public, health, and educational changes.

## 1. Introduction

Since the initial appearance of the coronavirus disease (COVID-19) caused by the novel coronavirus (SARS-CoV-2), and the beginning of the present worldwide pandemic as of March 2020 [1,2], the disease has, directly and indirectly, interfered with the general mental health of the global population. This primarily respiratory virus directly causes breathing disorders, which can cause confusion, depressed mood, impaired memory, anxiety, and post-traumatic stress disorder [3]. It even alters structures in the central nervous system that control sleep-wake cycles, resulting in an abnormal sleep rhythm [4], and it can cause hypoxic encephalopathy [3]. Recommended socio-epidemiological measures, such as protective mask requirements, isolation, limited contacts, and modified working conditions, along with the constant overload of media news related to COVID-19 and the lack of effective treatment or medications, all affect the mental health of the population [5]. The stigma towards those who are either sick with COVID-19 or have been in close contact with a sick person additionally affects their already impaired mental health [6]. Fear, as a multidimensional factor, can be one of the most significant elements that can lead to the impairment of mental health and well-being [7]. Previous studies have reported that psychological stress, insomnia, post-traumatic stress symptoms, depression, and anxiety are the most common manifestations of COVID-19-related fear, as well as a manifestation of mental fatigue caused by the quarantine, lockdown, and social distancing [8,9,10,11]. Stress affects one’s behavior such that it leaves a person unable to cope with the pressure, and as an attempt to relax and address stress more easily, a person often resorts to self-medication, the effects of which are unfortunately instant and temporary, leading to drug addiction [12].

The first case of a person infected by the SARS-CoV-2 virus, i.e., COVID-19, in the Republic of Serbia was recorded on 6 March 2020, followed by the state of emergency being declared on the territory of the Republic of Serbia on 15 March 2020. Schools and universities were temporarily closed, mass gatherings were banned, and a curfew was introduced three days later [13]. Educational activities were completely disrupted by the situation, and traditional teaching and learning practices were replaced by online forms, thus affecting hundreds of millions of students and teachers across the world [14]. Even though the outcomes of such a transition will be evaluated in the upcoming period, current studies have nevertheless promoted and highlighted the advantages of such drastic changes [15,16]. However, further studies are needed to assess the effects in terms of overall success in learning, mental health, and new skills acquisition and the correlations between online teaching methods and different fields of study. In the education of medical and nursing students, the online method is recognized as effective, but under the conditions of the COVID-19 pandemic, the psychological state of students can affect their satisfaction with online education [17]. Students’ anxiety related to COVID-19 had a negative role in the nursing students’ online learning [18].

In general, students’ satisfaction can be seen as a short-term attitude, which refers to the subjective students’ assessments of the extent to which their expectations have been met or exceeded in the given educational experience [19]. Bearing in mind that students form numerous expectations about their educational experiences, many authors have conceptualized students’ satisfaction as a multidimensional construct [20,21,22]. Academic aspects of satisfaction include some of the key factors, such as perceived quality of teaching, feedback provided by teachers, teaching styles, etc. The general characteristics of the courses, curricula, and teaching materials were also listed as significant. The subjective life experiences of students during their studies have been recognized as an important factor influencing students’ satisfaction [20,21,22].

Before the pandemic, numerous studies were conducted on online teaching to investigate students’ satisfaction and the factors affecting it [23,24]. However, the literature on students’ satisfaction regarding enforced online education during the COVID-19 pandemic is relatively scarce [25], although there is concrete evidence that variables of mental health, fear, and stress represent a major distraction and negatively influenced students’ satisfaction towards online learning [26,27].

On that account, our pilot study aimed to investigate how COVID-19-related stress and fear correlate with online teaching satisfaction among the nursing students at the Faculty of Medicine, University of Novi Sad, during the COVID-19 pandemic.

## 2. Materials and Methods

### 2.1. Research Sample and Setting

The usual, traditional method of teaching at the Faculty of Medicine was interrupted suddenly, without warning or preparation, in March 2020. Due to the pandemic, quarantine conditions, recommended socio-epidemiological measures, mandatory mask requirements, and physical distancing, practical and theoretical classes were shifted to an online model. In the new academic year of 2021/22, online learning became a common daily routine for all students. After almost two years, the beginning of the summer semester in February 2022 marked the return to classrooms and face-to-face learning.

This descriptive cross-sectional study was conducted from February to May 2022 among the nursing science students at the Faculty of Medicine, University of Novi Sad, Serbia, during the COVID-19 pandemic.

Students completed an anonymous online questionnaire on COVID-19-related stress, fear, and online teaching satisfaction via the Google Forms Administration App, which prevented multiple responses per email. All students involved in the study were informed about the study objectives, the data obtained and used in the study setting, anonymity of the data, instructions on how to fill out the questionnaire, and written informed consent. The minimum sample size computed for the nursing student population of the Faculty of Medicine of University of Novi Sad using $n^{\prime }=\frac{n}{1+\frac{z2xp(1-p)}{e2N}}$ (z—z score; e—margin of error, N—population size, p—population proportion) was 159 students (z = 159, e = 5%, N = 270). All study procedures were conducted in accordance with the Helsinki Declaration.

### 2.2. Instruments

The questionnaire consisted of three sections and was based on the COVID-19 Stress Scales (CSS) [28] and the Fear of COVID-19 scale (FCV-19S) [29]. The first section (CSS) focused on assessing COVID-19-related stress, the second (FCV-19S) assessed COVID-19-related fear, and the third part was concerned with online teaching satisfaction. For this study, socio-demographic data included age, sex, and year of study. Our research procedures included the following:A.To establish stress levels related to COVID-19, the COVID-19 Stress Scales (CSS), consisting of 36 items, were used [28]. Taylor et al. [28] summarized the psychological effects of COVID-19 under the name of COVID-19 Stress Syndrome, described in six subscales: 1. Danger subscale (DS)—worry about infection (items 1–6); 2. Socio-economic consequences subscale (SC)—worry about to the socio-economic consequences of the pandemic (items 7–12); 3. Xenophobia associated with the disease (XC)—fear that foreigners are potential carriers of the infection (items 13–16); 4. Contamination subscale (CS)—stress associated with contamination (items 19–24); 5. Traumatic stress symptoms associated with the pandemic subscale (TS) (e.g., nightmares, intrusive thoughts) (items 25–30); and 6. Compulsive checking and reassurance seeking subscale (CC) (items 31–36). The CSS is one of the few instruments assessing emotional reactions to the pandemic with a specific assessment of xenophobia as a factor contributing to fear and avoidance [28]. Items were rated on a 5-point Likert scale for the first four subscales, namely 0-not at all, 1-slightly, 2-moderately, 3-very, and 4-extremely; while in the fifth and sixth subscales, the items were rated with 0-never; 1-rarely; 2-sometimes; 3-often; and 4-almost always. The overall score across all six subscales varied from 0 to 144 with scores interpretated as follows: <5 very low COVID-19 stress, 5–16 low COVID-19 stress, 17–36 moderate COVID-19 stress, 37–71 high COVID-19 stress and >72 very high COVID-19 stress [12]. The Cronbach’s alpha coefficient of the CSS was 0.932 (n = 36), indicating a very high level of reliability [30]. The scale was validated and translated into Serbian in 2021 [31].B.COVID-19-related fear was measured by a one-dimensional COVID-19 Fear Scale (FCV-19S) [29], which uses seven items to assess the gravity of COVID-19-related fear. The participants indicated their level of agreement using a 5-item Likert-type scale as follows: 1-strongly disagree, 2-disagree, 3-neither agree or disagree, 4-agree and 5-strongly agree. The overall score ranged between 7 and 35 points with higher scores indicating greater COVID-19-related fear. The Cronbach’s alpha coefficient for the FCV-19S was α = 0.902 (n = 7), indicating very high internal consistency of the scale.C.For the assessment of online teaching satisfaction, the modified multidimensional Online Teaching Satisfaction Scale was used [32]. Its 36 items are divided into five subscales as follows: 1. Student Perception (SP) (9 items), 2. Lecturers’ subscale (LS) (9 items), 3. Technical Characteristics (TC) (5 items), 4. Management and Coordination of the lectures (MC) (4 items), and 5. Satisfaction (SS) (9 items). Each item was scored 1 to 5 points, as follows: 1-strongly disagree, 2-disagree, 3-neither agree or disagree, 4-agree, and 5-strongly agree. The overall score ranged from 36 to 180 with scores interpreted as follows: 36—unsatisfied, 37–72 slightly unsatisfied, 73–108 neutral, 109–144 satisfied, and >145 very satisfied. The Cronbach’s alpha coefficient of the satisfaction scale was 0.914 (n = 36), indicating a very high level of reliability [30].

### 2.3. Data Analysis

The collected data regarding COVID-19-related stress and fear and online teaching satisfaction were analyzed using descriptive statistics. The Statistical Package for Social Sciences (SPSS) IBM Statistics, version 26.0, was used to analyze the collected data. If the data followed a normal distribution, they were presented as the mean ± SD, and if they were not normally distributed, they were presented as the median (25th–75th percentiles). Pearson’s correlation coefficient was used for testing correlation between variables. The independent-samples *t*-test was used to compare continuous variables between two groups and one-way analysis of variance (ANOVA) to compare continuous variables among the four groups. Cronbach’s alpha coefficient was used to assess the internal consistency of the scales used in the study. A *p*-value less than 0.05 was considered statistically significant.

## 3. Results

### 3.1. Socio-Demographic Characteristics of the Sample

A total of 167 nursing students were included in the study, from the population of n = 270 with an estimated participation rate of 61.85%. The study participants were predominantly female 147 (88.0%), with a mean age of 21.3 ± 5.3. The distribution of students per year was as follows: 68 (41.2%) first-year students, 40 (24.0%) second-year students, 39 (23.4%) third-year students, and 20 (11.4%) fourth-year students.

### 3.2. COVID-19-Related Stress

Out of 167 students participating in the study, 5 (3.0%) had very low COVID-19 stress levels, 31 (18.6%) had low COVID-19 stress levels, 62 (37.1%) had moderate COVID-19 stress levels, 48 (28.7%) had high COVID-19 stress levels, and 21 (12.6%) had very high COVID-19 stress levels. When individual items of CSS were analyzed, the highest mean values were observed for the 26th statement —”Disturbing mental images about the virus popped into my mind against my will” (1.84 ± 1.10) —and the 36th statement — “Checked social media posts concerning COVID-19” (1.84 ± 1.18); and the lowest values were recorded for the 9th statement — “I am worried about grocery stores running out of cleaning or disinfectant supplies” (0.54 ± 0.92) —and the 27th statement — “I had trouble sleeping because I worried about the virus” (2.83 ± 1.20).

Table 1 shows descriptive data regarding the different CSS subscales. First-year and fourth-year health science students scored significantly lower overall (*p* < 0.001), as well as on the Xenophobia (*p* < 0.001) and Traumatic stress (*p* < 0.001) CCS subscales, compared to second-year and third-year students, while first-year students also scored significantly lower on the Danger subscale (*p* = 0.03) and Contamination subscale (*p* < 0.001). No difference was observed on the CSS regarding sex (*p* > 0.05).

One-factor analysis of variance was applied to examine the difference in COVID-19-related stress, which was indicated by the overall score on the scale and the scores on the individual subscales, according to the participants’ year of study. Four categories were formed: first, second, third, and fourth year of study. Table 2 shows the values of F-tests and the level of significance, as well as arithmetic means and standard deviations. It can be seen that the four examined groups showed statistically significant differences (*p* < 0.05) in the overall score and in all other scales except the Economic consequences domain. Subsequent comparisons using the post-hoc LSD test showed that first-year students achieved significantly lower results on the overall score than second-year and third-year students. Fourth-year students also had a lower score on the COVID-19 Stress Scale compared to third-year students. No statistically significant differences could be observed among the other groups in the overall score. In the Danger subscale, first-year students achieved significantly lower scores compared to all three remaining groups of students. There was no difference between the other groups regarding the Danger subscale. In the Xenophobia subscale, first-year students had lower scores than second-year and third-year students. Fourth-year students also had significantly lower scores compared to third-year students. In the Contamination subscale, first-year students had lower scores than second-year and third-year students. In the Traumatic stress and Compulsion subscales, first- and fourth-year students had lower scores compared to second-year and third-year students (Table 2).

### 3.3. COVID-19-Related Fear

One-factor analysis of variance was used to examine the difference in the scores achieved on the COVID-19-related fear scale among the participants according to the year of study. Table 3 below shows the F-test value and the significance level, as well as arithmetic means and standard deviations. A review of the table shows that the four examined groups differed in a statistically significant way (*p* < 0.05). Subsequent comparisons using the post-hoc LSD test showed that first-year students achieved significantly lower results on this scale compared to second-year and third-year students. No statistically significant differences were observed among the other groups.

Out of the whole sample, the overall FCV-19S score was 14.41 ± 5.88. First-year students experienced (*p* < 0.001) less COVID-19 fear compared to senior students. There was no difference observed on the FCV-19S regarding sex (*p* > 0.05). When individual items regarding the FCV-19S were analyzed, the highest mean values were observed for the second statement — “It makes me uncomfortable to think about COVID-19” (2.57 ± 1.25) —and the 5th statement “When watching news and stories about coronavirus-19 on social media, I become nervous or anxious” (2.49 ± 1.27); and the lowest for the 3rd statement — “My hands become clammy when I think about COVID-19” (1.76 ± 0.95) —and the 7th statement “My heart races or palpitates when I think about getting COVID-19” (1.78 ± 0.92).

### 3.4. COVID-19 Online Teaching Satisfaction Scale

The mean value of the overall scale was 132.56 ± 21.52, where the minimum value was 3, and the maximum value was 173. The mean value of the Students’ Perception subscale was 26.38 ± 6.91; for the Lecturers subscale, it was 36.88 ± 7.22; and it was 25.65 ± 3.82 for the Technical Characteristics subscale, 15.45 ± 2.35 for the Teaching Management subscale, and 29.24 ± 4.56 for the Satisfaction subscale.

There was no difference observed on the Online Teaching Satisfaction Scale regarding sex (*p* > 0.05). When individual items of the Online Teaching Satisfaction Scale were analyzed, the highest mean values were observed for the 32nd question — “How often during the winter semester were you determined and responsible in fulfilling your obligations when the situation required it?” (4.40 ± 0.78) —and the 33rd question — “During the winter semester, how often were you interested in studies?” (4.38 ± 0.74); and the lowest values for the 8th statement — “Online teaching effectively replaces the traditional way” (2.72 ± 1.32)—and 9th statement—“The advantages of online classes outweigh the disadvantages of canceling regular classes” (2.83 ± 1.20).

One-factor analysis of variance was used to examine the difference in Overall Satisfaction with online, teaching as well as across the subscales of this scale, according to the participants’ year of study. Table 4 shows that the four groups differed in a statistically significant way (*p* < 0.05) only in terms of the scores in the Students’ Perception domain. Subsequent comparisons using the post-hoc LSD test showed that first-year students achieved significantly lower scores in this domain compared to second-year and third-year students. Among the other groups, no statistically significant differences were observed in this domain.

### 3.5. COVID-19-Related Stress and Fear in Correlation with Online Teaching

Based on the results of Table 4, the COVID-19 Stress Scales and the COVID-19 Fear Scale revealed a significant, high correlation (r = 0.68; *p* < 0.001). The analysis of the Online Teaching Satisfaction Scale in relation to the COVID-19 Stress Scales and the COVID-19 Fear Scale revealed several significant correlations. The overall score on the Online Teaching Satisfaction Scale positively and moderately correlated with the overall scores on the Fear of COVID-19 Scale (r = 0.18; *p* < 0.05) and the COVID-19 Stress Scales (r = 0.19; *p* < 0.05). There were also moderate, positive correlations of the overall score on the Online Teaching Satisfaction Scale with the Economic Consequences (r = 0.16; *p* < 0.05), Xenophobia (r = 0.15; *p* < 0.05), and Contamination subscales (r = 0.17; *p* < 0.05). The analysis of the subscales of the Online Teaching Satisfaction Scale showed that the subscales of Students’ Perception (r = 0.25; *p* < 0.01) and Lecturers significantly correlated with the overall scores on the COVID-19 Fear Scale (r = 0.21; *p* < 0.01) and the COVID-19 Stress Scale (r = 0.24; *p* < 0.01), as well as with all scores on the subscales of the COVID-19 Stress Scale. Other correlations were not statistically significant (Table 5). 

## 4. Discussion

To the best of our knowledge, this study is one of the first in the West Balkans investigating the impact of the COVID-19 pandemic on the student population, as well as the cohabitations of COVID-19-related stress, fear, and online teaching satisfaction among students during the pandemic. The study sample included participants who were mostly female, young, and first-year students and who experienced moderate to very high COVID-19-related stress, experienced COVID-19-related fear, and were relatively satisfied with online teaching during the COVID-19 pandemic.

Our research results confirmed the results of previous studies among medical science students in the EU member countries, which recorded an increase in stress and anxiety by about 60% among students in France, Spain, and Poland due to COVID-19 [33,34,35]. Increased mental stress, feelings of loneliness, and fear of the future were also reported by students in Germany, whereas an increase in COVID-19-related depression and anxiety was recorded in Switzerland [36,37,38]. Moreover, the different prevalence of COVID-19 fear across different countries can be attributed to cultural and other contextual factors and differences in medical care and mental health care accessibility [39]. For example, the requirement to wear masks, an effective measure to combat airborne pathogens, including SARS-CoV-2, is often seen as a trigger for cultural conflicts and is even considered to be cultural violence [40].

In our study, first-year and fourth-year nursing students scored lower on the subscales of illness-related Xenophobia, Traumatic stress and Compulsive checks on the CSS compared to second-year and third-year students. The youngest participants in our study, first-year students, also scored significantly lower than all three remaining groups on the Danger subscale. They also scored lower on the Online Teaching Satisfaction Scale and experienced less fear of COVID-19. Our study found that younger nursing students had less fear and stress, probably as a result of their new study environment and unfamiliarity with the traditional teaching practices of the faculty. Studies conducted during the pandemic revealed strong links between sex and COVID-19-related fear and anxiety. Numerous studies have reported higher levels of COVID-19-related fear and the perception of danger among women [39], as well as higher anxiety in women [41,42]. On the other hand, some studies showed that men have higher levels of anxiety due to COVID-19 [43,44]. In addition, some authors found no sex differences in fear and anxiety related to COVID-19, consistent with the results of our study [45,46,47].

The study also showed that the higher intensity of COVID-19-related fear corresponds to higher levels of COVID-19-related stress. This finding is not surprising since COVID-19-related fear and COVID-19-related stress have been found to be linked in a kind of caught in a vicious circle of anxiety and depression, which together lead to a deterioration of students’ mental health at multiple levels [48].

On the one hand, fear is prevalent among our students as a protective measure, which helps them to limit their exposure to infected individuals and follow epidemiological measures, such as social distancing, wearing face masks properly, and washing hands. On the other hand, it is also one of the reasons for the stigmatization of the infected or those who have been in close contact with them, leading to further social isolation, mental health impairment, neglecting of COVID-19 symptoms, postponing of adequate treatment recommended in the early stages of the disease, and further spreading of COVID-19 and subsequent mental health issues [48,49].

High COVID-19-related stress among students may be associated with changes in sleep and diet, substance abuse, and worsening of chronic illnesses and may affect socioeconomic stability, all of which impair mental health and worsen it [49,50,51]. It might also be linked to nursing students being involved in the fight against the disease with increased anxiety and fear of terrible outcomes and mortality among infected people. They were also in close contact with severely sick patients in the health care institutions where they sought treatment, and they were also exposed to persistent media pressure with COVID-19-related news (death tolls, severe cases, and overcrowded health systems) [49,50].

Fear represents an emotional state that causes panic, social isolation, or loss of quality of life that potentially hinders individual performance [47,50]. Some have authors stated that the prolonged closure of educational institutions causes fear among the majority of students regarding the loss of educational achievement, as well as fear of COVID-19 encouraging loss of interest and students’ withdrawal from educational activities [47,50].

One possible reason for COVID-19-related stress among nursing students is the sudden transition from the traditional class-based teaching process to the novel online teaching process. Overall, our students were relatively satisfied with the online teaching process, which proved to have numerous advantages, such as remote learning, comfort, and accessibility [16]. However, some studies emphasized the disadvantages of such a setting for practical sessions and clinical practical training [51], accompanied by long screen times, inefficiency, and difficulty in maintaining academic integrity [16,51]. These results support the findings of Zolotov et al. (2020), in which students were found to have conflicting opinions about satisfaction with online classes [47]. Some students experienced the online method as a significant learning opportunity, while others viewed it as a poor alternative to traditional learning. The analysis of students’ satisfaction with online teaching showed that first-year students achieved significantly lower results in the Student Perception subscale compared to the other groups. This group of students, unlike the others, did not even have the opportunity for traditional learning at the university. Therefore, we can conclude that, for most students, especially those who were learning online for the first time during the COVID-19 pandemic, it is a challenge. However, for individual students, various online learning modules offer experiential and constructive learning environments.

In our study, differences in the self-assessment of students’ satisfaction with online teaching were determined from the aspect of students’ subjective experiences. In particular, no differences were observed between the groups in the domain of academic aspects that included teaching characteristics, satisfaction with feedback during online classes, communication with lecturers and colleagues, teaching styles, etc., as well as in the domain of general attributive characteristics of the courses that they attended. A possible explanation can be found in first-year students feeling a lack of support from peers and resources, and they felt isolated. In our study, we found that moderate anxiety proved beneficial for online learning satisfaction.

### 4.1. Implications

The findings of this study showed that the nursing student population is at risk for developing mental illnesses due to COVID-19-related fear and stress. Coping with and recognizing potential mental health issues require a high level of attention from both nursing students and their teachers. Understanding and accepting these findings will enable the design of adaptive and acceptable peer support strategies. Summarizing the satisfaction of students with online teaching during the COVID-19 pandemic, the obtained results represent the basis for introducing necessary changes both in course curricula and in the innovative methodological approaches of teachers. A stratified program is needed to prevent further decline of students’ mental health and to improve their adaptation through public health and educational changes.

### 4.2. Limitations

This study had several limitations. First, our study sample lacked information about participants’ previous psychiatric and somatic profiles, which might stimulate and exacerbate the development of fear, stress, and other mental health challenges. Further studies should incorporate an extensive psychiatric interview to establish the existing individual mental health status and the presence of anxiety, depression, phobias, and other mental health issues. Second, the cross-sectional study setting prevented the inference of causality. Third, the sampling method of this pilot study included only nursing students and a small sample size. Further studies should include students from diverse majors, a larger sample, and possibly different universities in Serbia to calculate differences between regions in the country.

In this pilot study, the assessment of online teaching satisfaction relied on only three dimensions that described the phenomenon of satisfaction. Further studies should include more variables, such as self-efficacy and resilience in relation to fear and stress due to COVID-19.

## 5. Conclusions

In conclusion, nursing students in Serbia experienced moderate to very high COVID-19-related stress and fear of COVID-19 and were relatively satisfied with online teaching during the COVID-19 pandemic. The higher intensity of COVID-19-related fear was observed to be linked to higher levels of COVID-19-related stress. Even though the online teaching setting was well tolerated by health care students, the present frequency of mental health problems, such as COVID-19-related fear and stress, is worrying and necessitates prompt engagement of health organizations and service commissioners to prevent a mental health disaster, which could be further worsened not only by the pandemic also but by the post-pandemic global, financial, and social situation.

## Figures and Tables

**Table 1 healthcare-11-00894-t001:** Descriptive data regarding different COVID-19 stress scale subscales.

		Min	Max	M	SD	Skewness	Kurtosis
COVID-19 Stress Scales (CSS)	CSS overall	1	102	36.51	24.28	0.66	−0.49
	Subscale						
	Danger	0	21	7.56	4.99	0.22	−0.77
	Socio-economic consequences	0	21	3.87	5.19	1.33	0.79
	Xenophobia	0	22	4.88	5.42	1.09	0.32
	Contamination	0	20	5.05	5.20	1.01	0.25
	Traumatic stress	0	22	6.47	5.41	0.68	−0.59
	Compulsive checking	0	24	8.65	5.83	0.78	−0.24

**Table 2 healthcare-11-00894-t002:** Differences in COVID-19 Stress Scale according to the year of study.

	Year of Study	Mean	SD	F	df	*p*
CSS overall score	First	26.15	20.27	10.54	3 (163)	0.00
	Second	44.05	25.28			
	Third	48.89	24.45			
	Fourth	32.84	18.82			
Subscale						
Danger	First	6.26	4.60	2.94	3 (163)	0.03
	Second	8.40	5.20			
	Third	8.20	5.32			
	Fourth	9.21	4.44			
Socio-economic consequences	First	3.23	4.93	0.65	3 (163)	0.58
	Second	4.25	5.31			
	Third	4.56	5.67			
	Fourth	4.00	4.81			
Xenophobia	First	3.24	4.58	6.86	3 (163)	0.00
	Second	5.80	5.83			
	Third	7.58	5.98			
	Fourth	3.36	3.49			
Contamination	First	3.02	4.08	7,78	3 (163)	0.00
	Second	7.32	6.17			
	Third	6.43	5.04			
	Fourth	4.84	4.42			
Traumatic stress	First	3.85	3.92	15.03	3 (163)	0.00
	Second	8.45	5.58			
	Third	9.66	5.72			
	Fourth	5.26	4.09			
Compulsive checking	First	6.53	4.41	12.25	3 (163)	0.00
	Second	9.82	6.13			
	Third	12.49	6.13			
	Fourth	6.15	4.66			

**Table 3 healthcare-11-00894-t003:** Differences in COVID-19-related fear according to the year of study.

	Year of Study	Mean	SD	F	df	*p*
Fear of COVID-19 scale (FCV-19S)	First	12.40	5.28	5.34	3 (163)	0.00
	Second	16.39	5.62			
	Third	16.00	6.39			
	Fourth	14.16	5.34			

**Table 4 healthcare-11-00894-t004:** Differences in online teaching satisfaction according to the year of study.

	Year of Study	Mean	SD	F	df	*p*
Online Teaching Satisfaction Scale overall	First	129.07	24.03	1.11	3 (163)	0.34
	Second	135.82	21.95			
	Third	135.25	18.37			
	Fourth	132.84	15.80			
Subscale						
Student perception	First	24.92	7.29	3.08	3 (163)	0.02
	Second	27.70	7.62			
	Third	28.41	5.86			
	Fourth	24.73	4.24			
Lecturers’ subscale	First	36.20	6.94	0.47	3 (163)	0.70
	Second	37.85	8.77			
	Third	37.23	5.82			
	Fourth	36.57	7.47			
Technical characteristics	First	25.49	2.92	1.05	3 (163)	0.37
	Second	26.42	4.88			
	Third	24.97	4.29			
	Fourth	26.00	2.92			
Management and coordination of the lectures	First	15.50	1.95	0.66	3 (163)	0.57
	Second	15.02	2.53			
	Third	15.74	2.68			
	Fourth	15.57	2.56			
Satisfaction	First	29.50	4.20	0.41	3 (163)	0.78
	Second	28.82	5.22			
	Third	28.89	4.61			
	Fourth	29.94	4.45			

**Table 5 healthcare-11-00894-t005:** Correlation of the scales and subscales used in the research.

Scale	FCV-19S	CSS Overall	DS	SE	XS	CS	TS	CC	OTS Overall	SP	LS	TC	MC	SS
**FCV-19S**	-													
**CSS overall**	0.68 **	-												
DS	0.54 **	0.73 **	-											
SE	0.53 **	0.74 **	0.58 **	-										
XS	0.55 **	0.80 **	0.48 **	0.60 **	-									
CS	0.60 **	0.84 **	0.51 **	0.56 **	0.73 **	-								
TS	0.64 **	0.84 **	0.55 **	0.48 **	0.56 **	0.68 **	-							
CC	0.26 **	0.58 **	0.23 **	0.17 *	0.29 **	0.37 **	0.52 **	-						
**OTS overall**	0.18 *	0.19 *	0.11	0.16 *	0.15 *	0.17 *	0.13	0.12	-					
SP	0.25 **	0.32 **	0.17 *	0.22 **	0.24 **	0.26 **	0.31 **	0.25 **	0.78 **	-				
LS	0.21 **	0.24 **	0.14 *	0.15 *	0.18 *	0.26 **	0.24 **	0.15*	0.84 **	0.69 **	-			
TC	0.08	0.04	0.08	0.10	0.00	0.00	0.01	0.01	0.71 **	0.43 **	0.51 **	-		
MC	0.09	0.04	0.04	0.05	0.06	0.04	0.05	−0.03	0.61 **	0.28 **	0.47 **	0.42 **	-	
SS	−0.12	−0.11	0.00	−0.06	0.00	−0.07	−0.04 **	−0.12	0.47 **	0.11	0.15 *	0.33 **	0.38 **	-

** *p* < 0.01, * *p* < 0.05.

## Data Availability

The data are available from the authors on personal demand.

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
