# Peer review of "COVID-19-Related Stress, Fear and Online Teaching Satisfaction among Nursing Students during the COVID-19 Pandemic"

_healthcare, 2023, doi:10.3390/healthcare11060894_

Round 1
Reviewer 1 Report
This study is meaningful in that this research is one of the first studies in the West Balkan region examining the impact of the COVID-19 pandemic on the student population, as well as the cohabitation of COVID-19 related stress, anxiety, and satisfaction with online teaching among students during pandemic periods. However, there are some parts which should be revised before publication.
First of all, abstract seems to be too complicated and specific with lots of numbers and info. Thus, it would be required to make it concise and compact.
In the methodology section, I guess it would be better to provide a little more detailed information on the research context so that readers can understand the contextual situation of the research. Now it only has information on the participants and their faculty.
All the tables in the manuscript must be revised and formatted according to the guidelines of the journal submission. At its current form, it is very hard to focus on the contents of the manuscript. In addition, it would be required to check and revise all the stylistic errors (I guess there are quite lots of errors) in the manuscript.
Lastly, this study provide lots of findings and results, but it doesn’t seem to provide enough academic and practical implications based on the findings. So it would be better to provide implications for future research, teaching in the field, and so on.
Author Response
Dear Editor,
Thank you for giving us the opportunity to resubmit our manuscript 'COVID-19-related Stress, Fear and Online Teaching Satisfaction Among Nursing Students During the COVID-19 Pandemic'. We really appreciate the time and effort that you and the reviewers have dedicated to providing your valuable feedback on our manuscript. We are grateful to the reviewers for their insightful comments on our paper. We have tried to incorporate changes to reflect most of the suggestions provided by the reviewers.
Here is a point-by-point reply to the reviewers’ comments and concerns.
All changes in the manuscript are marked in yellow.
Comments from reviewer 1:
|
|
Comment: |
Answer: |
|
1. |
This study is meaningful in that this research is one of the first studies in the West Balkan region examining the impact of the COVID-19 pandemic on the student population, as well as the cohabitation of COVID-19 related stress, anxiety, and satisfaction with online teaching among students during pandemic periods. However, there are some parts which should be revised before publication. |
Thank you for recognizing the importance of our study. |
|
2. |
First of all, abstract seems to be too complicated and specific with lots of numbers and info. Thus, it would be required to make it concise and compact. |
The abstract was corrected in accordance with the reviewer's suggestions. |
|
3 |
In the methodology section, I guess it would be better to provide a little more detailed information on the research context so that readers can understand the contextual situation of the research. Now it only has information on the participants and their faculty. |
We have corrected everything that was suggested. |
|
4. |
All the tables in the manuscript must be revised and formatted according to the guidelines of the journal submission. At its current form, it is very hard to focus on the contents of the manuscript. In addition, it would be required to check and revise all the stylistic errors (I guess there are quite lots of errors) in the manuscript. |
All the tables have been formatted according to the guidelines of the journal. |
|
5. |
Lastly, this study provide lots of findings and results, but it doesn’t seem to provide enough academic and practical implications based on the findings. So it would be better to provide implications for future research, teaching in the field, and so on. |
We thank the reviewer for this comment. The practical implications have been added in the manuscript. |
Reviewer 2 Report
Respected
I consider the topic very important and interesting, and I believe that this work would contribute significantly to the journal.
I propose a major revision of the work in relation to the following items:
Title
The title is extremely long. I believe that Covid-19 stress, fear and satisfaction of online teaching among nursing students during covid 19 pandemic would be more appropriate
Abstract
In the abstract, numerous spelling mistakes were made and the sentences do not have a clear beginning and end, it is necessary to correct this. It is also not necessary to burden the abstract with so many numerical data.
Introduction
I think it is clearly and concisely written
Methodology
The methodology of the work is written with high quality.
In the part of statistical analysis, the correlation coefficient is missing and the phi square test was added, which was not used
Results
The description of individual items in table 2, 3 and 4 is completely unnecessary and burdens the work. If the authors recognize the importance of displaying results on individual items, they can leave that information in the text.
Instead of individual items, it would be useful to see the scores on the COVID fear scale by year as well as the results of the F test in a table. The same applies to satisfaction with online teaching. Extract individual scores and display scores by age in relation to subscales and F test.
Discussion
Correctly written
I encourage the authors to correct their work and to persevere in publishing this interesting scientific work
Sincerely
Author Response
Dear Editor,
Thank you for giving us the opportunity to resubmit our manuscript 'COVID-19-related Stress, Fear and Online Teaching Satisfaction Among Nursing Students During the COVID-19 Pandemic'. We really appreciate the time and effort that you and the reviewers have dedicated to providing your valuable feedback on our manuscript. We are grateful to the reviewers for their insightful comments on our paper. We have tried to incorporate changes to reflect most of the suggestions provided by the reviewers.
Here is a point-by-point reply to the reviewers’ comments and concerns.
All changes in the manuscript are marked in yellow.
Comments from reviewer 2:
|
|
Comment: |
Answer: |
|
1 |
I consider the topic very important and interesting, and I believe that this work would contribute significantly to the journal. |
Thank you for your comment and for recognizing the importance of the research topic. |
|
2 |
Title The title is extremely long. I believe that Covid-19 stress, fear and satisfaction of online teaching among nursing students during covid 19 pandemic would be more appropriate |
Thank you for pointing this out. We have modified the title in accordance with your proposal.
|
|
3 |
Abstract In the abstract, numerous spelling mistakes were made and the sentences do not have a clear beginning and end, it is necessary to correct this. It is also not necessary to burden the abstract with so many numerical data. |
The manuscript has been proofread. |
|
4 |
Introduction I think it is clearly and concisely written |
Thank you for your comment. |
|
5 |
Methodology The methodology of the work is written with high quality.
|
Thank you for your comment. |
|
6 |
In the part of statistical analysis, the correlation coefficient is missing and the phi square test was added, which was not used |
We have excluded Phi square test and added the Pearson correlation coefficient
|
|
7 |
Results The description of individual items in table 2, 3 and 4 is completely unnecessary and burdens the work. If the authors recognize the importance of displaying results on individual items, they can leave that information in the text. Instead of individual items, it would be useful to see the scores on the COVID fear scale by year as well as the results of the F test in a table. The same applies to satisfaction with online teaching. Extract individual scores and display scores by age in relation to subscales and F test. |
We have excluded Tables 2, 3, and 4.The results of individual items are kept in the text.We presented the results of the F test according to the year of study: Table 2 Covid-19 stress scale; Table 3 Covid fear; Table 4 Online satisfaction. |
|
8 |
Discussion Correctly written I encourage the authors to correct their work and to persevere in publishing this interesting scientific work |
Thank you for your comment. |
Reviewer 3 Report
The present study is interested in dissecting the relationship between COVID-19 fear variables and online lectures. Although the pandemic is starting to be less prevalent in media coverage, it is important to now direct research to work through how things were handled and to discover how they should change for the future. As such, this paper deals with an interesting and important field of study. This being said, there are many shortcomings in this paper and there are major revisions needed should it be considered for publication:
· There are some language mistakes, which should be rectified. I urge the authors to let the paper proof-read by an English native speaker. Some of them are typos, like in the abstract on line 10, where after “COVID-19 fear”, there should be a comma instead of a dot, or line 81, where there should not be a dot before the comma. These things automatically leave the impression that the paper and the study were performed ‘in a rash’ and not done with the necessary care for quality. Please be aware about this for the future: having a well formulated paper without unnecessary typos (which can be spotted immediately and easily by proofreading it) increases trust in the content of the study.
· The abstract does not quite inform the readers about the purpose of this study. The authors seem to be interested in mental health impacts of COVID-19. For this reason, they implemented several scales (CSS, FCV-19S). However, they also assessed items concerning online teaching. Somehow, this appeared to be important to the study design. The “why” and “how” to this fact should be indicated much clearer in the abstract.
· On line 14, please insert the parentheses before the end of the sentence, before the dot.
· According to APA, p-values should be reported without a zero before the decimal points since everybody knows that the values can only be between zero and one. So, the initial zero is somewhat redundant. Hence, please change this wherever necessary: so, p<.05 instead of p<0.05. I know that MDPI editors do not seem to be concerned with this, but at least according to standard referencing guidelines 8 (like APA), this is the official recommendation.
· On line 36, the statement “Not only through the direct mechanisms…” needs further clarification.
· On line 41, before the word “which”, there is a comma missing. I will not make any more such remarks in the review – please hand the paper to an English proficient proof reader before resubmitting.
· On line 42, the statement “existence of the being is fear and stress” is philosophically dubious. Please change it or add clarification. First, the existence of being sounds like a redundancy. Second, the idea that “being is fear and stress” does not make much sense. Maybe the authors simply wanted to say that life in general can become stressful. But this is not how they put it here.
· The introduction needs to make much clearer what the current study is about. At the moment, it is extremely vague. The authors hold that, “further studies are needed”, which is fair enough, and then adding that, “The aim of our pilot-study was to analyze COVID-19 related fear, stress and satisfaction with online teaching among nursing science students…”. So what exactly are you trying to find out? Do you want to see if there is an association between fear and success of online education? If this is the case, the please clearly and deliberately say so. The intent of the study needs to be succinctly and evidently stated. And if so, what exactly is understood by the ‘success of online’ education? How happy students are with online teaching? Or if the students’ grades are better or worse after the online intervention? As it is written right now, the introduction gives an insufficient account of what the study’s goal is about.
· At the end of the introduction, the authors hold that the findings should “alert governmental bodies to an increasing mental health problem among students in Serbia.” This is not valid as an ex-ante presupposition. Maybe you can say ‘after the fact’ of the study, that now you want to alter the government of what is happening. But at this stage (in the introduction), you explain to the reader the goals of the study before it has occurred – and hence, if you assume that there are increasing mental health problems, you already make a statement that you could only know ‘after’ the study has been effected. Hence, this is a statement that you can put in the discussion or conclusion but not in the introduction. Otherwise, it leaves an extremely dubious impression since you would have biases that apparently you simply wanted to corroborate in this study.
· The methods section lacks the relevant citations. For example, of how to calculate the minimum sample size or the Helsinki declaration.
· The subsection “Data collection and study questionnaire” gives a good and detailed account of the items being used. Well done.
· On line 110, do you mean the “collection of data”?
· In the methodological section where the statistical analysis is discussed, the wording concerning the T-test and ANOVA is somewhat confusing. As a result, it is not quite clear if the authors are certain about what they are doing. Both the t-test as well as the ANOVA are parametric tests that are useful for determining statistical differences between continuous variables. T-tests can be used if one wants to discern differences between two variables and ANOVAs are useful if there are more variables (then the individual variance per item is compared to the grand mean of all the variables together, so to speak). I assume that the authors know this, but why did they write that they used the t-tests to “compare continuous variables” and ANOVAs “for parametric data”? Both are parametric since both are continuous and should be normally distributed. Consequently, then, chi-square was used for categorical variables. Please change the wording or further explain in regard to the t-test and the ANOVA.
· On line 132 and in table 1, one item is labelled “xenophobia”. Xenophobia is usually understood as a person’s aversion against foreigners or strangers. How is this relevant to the present study objectives? Please leave it out or explain in the manuscript.
· On line 137, the quotation marks are inserted incorrectly.
· Table 1 looks nicely done. However, COVID-19 is scattered on two lines, which is not good. Maybe you can simply put COVID and then put it on one single line. Tables 2 to 5 do not have the same format as table 1. They look like rapid copy-pastes. Please improve them so that they look like table 1. These are the things that really make reviewers question the quality of the whole paper (I urge authors to be more careful in their next submission – they can lead to an unsympathetic reading and faster rejections!)
· Table 2 to 5 take up a lot of space. Could they be inserted in a more condensed fashion? Or maybe as graphs? This is not a revision requirement but more of a friendly suggestion.
· As a reader, one is confronted with a lot of tables and numbers in the results section. As for the results chapter, it is always a trade-off between providing enough and sufficient information so that every reader has all the information needed to follow the discussion and to make up one’s own opinion (and maybe recalculate the results), but too much information that is not directly involved in answering the research questions makes it unreadable and distracting. Hence, I ask the authors to go through the results section once again and to decide upon whether all the information provided is strictly speaking necessary for answering their research questions. If it is, then keep it in the manuscript. For the parts that are “nice to have but not necessary”, move them to the supplementary material.
· This point is especially manifest in table 5. Here we find 14 items and a correlation matrix. First, it is not clear from the table alone which variables we are dealing with (what is variable 1, 2, 3, and so forth? There are some notes below the table, it is difficult to take in) – this should be visible from the table itself. Second, there is too much unnecessary information here. On the one hand, not all items seem to be relevant for the authors’ research questions. And on the other hand, also non-significant results are displayed. For most purposes, this is not necessary. Here, for example, you could just make a simple and small table depicting the relevant results of the correlation analysis.
· The results section itself – this is a huge concern – is insufficient and does not correspond to the methods section. In the methods section you should say that you want to do t-tests, ANOVAs and chi-square tests. Where do we find them in the results section? In the methods section, you should say that in order to answer the research question, we want to do A, B, and C. Then in the results section, you should report the findings of A, B, and C. This is not what we see here. At the end, we even find a correlation matrix, which was never introduced in the methods section. So there are many questions emerging, like: is this a Pearson matrix? Or did they use Kendall’s Tau or Spearman’s Rho?
· The discussion section is insufficient as well. You first need to quickly summarize what you did and what you wanted to find out. Then you should discuss your own findings and explain how they answered (or did not answer) your research questions. Consequently, you should then discuss how your findings correspond to the literature. Finally, some practical consequences can be mentioned.
· I suggest inserting a conclusion chapter at the end and also moving the limitations in this section.
· The perhaps biggest methodological shortcoming of this study is philosophical in nature. It has to do with the question what it is all about. The authors looked at variables concerning covid-19 stress and fear, eventually asking if online education is affected by this as well. This was predominantly analyzed via asking the respondents about their contentment with online education. However, looking at the effectiveness of online teaching is not necessarily measured best by simply asking students whether they are happy with the teaching mode. This is only one possible (albeit limited) approach to the question. The authors need to put in a lot more work convincing the readers that what they do actually corresponds to the validity of the study’s objectives.
Overall, this paper has major problems that are both methodological and redactional in nature. They are rooted very deep in the article. However, there is a lot of valuable data that the authors have worked on, and I believe that if the paper is thoroughly worked through once again and if considerable major revisions are applied (restructuring, redaction, language editing, and methodological strengthening of the arguments), the paper could be reconsidered for publication.
Author Response
Dear Editor,
Thank you for giving us the opportunity to resubmit our manuscript 'COVID-19-related Stress, Fear and Online Teaching Satisfaction Among Nursing Students During the COVID-19 Pandemic'. We really appreciate the time and effort that you and the reviewers have dedicated to providing your valuable feedback on our manuscript. We are grateful to the reviewers for their insightful comments on our paper. We have tried to incorporate changes to reflect most of the suggestions provided by the reviewers.
Here is a point-by-point reply to the reviewers’ comments and concerns.
All changes in the manuscript are marked in yellow.
Comments from reviewer 3:
|
|
Comment: |
Answer: |
|
1 |
There are some language mistakes, which should be rectified. I urge the authors to let the paper proof-read by an English native speaker. Some of them are typos, like in the abstract on line 10, where after “COVID-19 fear”, there should be a comma instead of a dot, or line 81, where there should not be a dot before the comma. These things automatically leave the impression that the paper and the study were performed ‘in a rash’ and not done with the necessary care for quality. Please be aware about this for the future: having a well formulated paper without unnecessary typos (which can be spotted immediately and easily by proofreading it) increases trust in the content of the study. |
We thank the reviewer for this comment. The language mistakes have been corrected by English proficient proofreader. |
|
2 |
The abstract does not quite inform the readers about the purpose of this study. The authors seem to be interested in mental health impacts of COVID-19. For this reason, they implemented several scales (CSS, FCV-19S). However, they also assessed items concerning online teaching. Somehow, this appeared to be important to the study design. The “why” and “how” to this fact should be indicated much clearer in the abstract. |
We have modified the abstract according to your suggestions. |
|
3 |
On line 14, please insert the parentheses before the end of the sentence, before the dot. |
The sentence on line 14 has been reworded. |
|
4 |
According to APA, p-values should be reported without a zero before the decimal points since everybody knows that the values can only be between zero and one. So, the initial zero is somewhat redundant. Hence, please change this wherever necessary: so, p<.05 instead of p<0.05. I know that MDPI editors do not seem to be concerned with this, but at least according to standard referencing guidelines 8 (like APA), this is the official recommendation. |
Corrected. |
|
5 and 6 |
On line 36, the statement “Not only through the direct mechanisms…” needs further clarification. On line 41, before the word “which”, there is a comma missing. I will not make any more such remarks in the review – please hand the paper to an English proficient proof reader before resubmitting. On line 42, the statement “existence of the being is fear and stress” is philosophically dubious. Please change it or add clarification. First, the existence of being sounds like a redundancy. Second, the idea that “being is fear and stress” does not make much sense. Maybe the authors simply wanted to say that life in general can become stressful. But this is not how they put it here. |
The sentences on lines 36, 41, and 42 have been reworded.
|
|
7 |
The introduction needs to make much clearer what the current study is about. At the moment, it is extremely vague. The authors hold that, “further studies are needed”, which is fair enough, and then adding that, “The aim of our pilot-study was to analyze COVID-19 related fear, stress and satisfaction with online teaching among nursing science students…”. So what exactly are you trying to find out? Do you want to see if there is an association between fear and success of online education? If this is the case, the please clearly and deliberately say so. The intent of the study needs to be succinctly and evidently stated. And if so, what exactly is understood by the ‘success of online’ education? How happy students are with online teaching? Or if the students’ grades are better or worse after the online intervention? As it is written right now, the introduction gives an insufficient account of what the study’s goal is about. |
Thank you very much for your comments. We have reworded the introduction according to your suggestions. |
|
8 |
At the end of the introduction, the authors hold that the findings should “alert governmental bodies to an increasing mental health problem among students in Serbia.” This is not valid as an ex-ante presupposition. Maybe you can say ‘after the fact’ of the study, that now you want to alter the government of what is happening. But at this stage (in the introduction), you explain to the reader the goals of the study before it has occurred – and hence, if you assume that there are increasing mental health problems, you already make a statement that you could only know ‘after’ the study has been effected. Hence, this is a statement that you can put in the discussion or conclusion but not in the introduction. Otherwise, it leaves an extremely dubious impression since you would have biases that apparently you simply wanted to corroborate in this study. |
Thank you for this comment. We excluded this sentence from the introduction. |
|
9 |
The methods section lacks the relevant citations. For example, of how to calculate the minimum sample size or the Helsinki declaration. |
The minimum sample size and the Helsinki Declaration are in the text.
|
|
10 |
The subsection “Data collection and study questionnaire” gives a good and detailed account of the items being used. Well done. |
Thank you for this comment. |
|
11 |
On line 110, do you mean the “collection of data”? |
Corrected. |
|
12 |
In the methodological section where the statistical analysis is discussed, the wording concerning the T-test and ANOVA is somewhat confusing. As a result, it is not quite clear if the authors are certain about what they are doing. Both the t-test as well as the ANOVA are parametric tests that are useful for determining statistical differences between continuous variables. T-tests can be used if one wants to discern differences between two variables and ANOVAs are useful if there are more variables (then the individual variance per item is compared to the grand mean of all the variables together, so to speak). I assume that the authors know this, but why did they write that they used the t-tests to “compare continuous variables” and ANOVAs “for parametric data”? Both are parametric since both are continuous and should be normally distributed. Consequently, then, chi-square was used for categorical variables. Please change the wording or further explain in regard to the t-test and the ANOVA. |
Thank you for this comment. We have inserted the correction according to your suggestions. |
|
13 |
On line 132 and in table 1, one item is labelled “xenophobia”. Xenophobia is usually understood as a person’s aversion against foreigners or strangers. How is this relevant to the present study objectives? Please leave it out or explain in the manuscript. |
In accordance with the standardized scale (Taylor et al., 2020), the COVID-19 Stress Scale describes the COVID-19 stress syndrome through six subscales (explained in detail in the 2. Materials and Methods section, part 2.2. Instruments). One of the subscales explains the anxiety about foreigners who could be carriers of the infection (items 13–16). It was named Xenophobia subscale because it was related to the disease. |
|
14 |
On line 137, the quotation marks are inserted incorrectly. |
Corrected. |
|
15 |
Table 1 looks nicely done. However, COVID-19 is scattered on two lines, which is not good. Maybe you can simply put COVID and then put it on one single line. Tables 2 to 5 do not have the same format as table 1. They look like rapid copy-pastes. Please improve them so that they look like table 1. These are the things that really make reviewers question the quality of the whole paper (I urge authors to be more careful in their next submission – they can lead to an unsympathetic reading and faster rejections!) |
Corrected. |
|
16 |
Table 2 to 5 take up a lot of space. Could they be inserted in a more condensed fashion? Or maybe as graphs? This is not a revision requirement but more of a friendly suggestion. |
Thank you very much for your comment. We have taken your friendly advice and removed Tables from 2 to 5. |
|
17 |
As a reader, one is confronted with a lot of tables and numbers in the results section. As for the results chapter, it is always a trade-off between providing enough and sufficient information so that every reader has all the information needed to follow the discussion and to make up one’s own opinion (and maybe recalculate the results), but too much information that is not directly involved in answering the research questions makes it unreadable and distracting. Hence, I ask the authors to go through the results section once again and to decide upon whether all the information provided is strictly speaking necessary for answering their research questions. If it is, then keep it in the manuscript. For the parts that are “nice to have but not necessary”, move them to the supplementary material. |
Thank you for your comment. Corrected. |
|
18 |
This point is especially manifest in table 5. Here we find 14 items and a correlation matrix. First, it is not clear from the table alone which variables we are dealing with (what is variable 1, 2, 3, and so forth? There are some notes below the table, it is difficult to take in) – this should be visible from the table itself. Second, there is too much unnecessary information here. On the one hand, not all items seem to be relevant for the authors’ research questions. And on the other hand, also non-significant results are displayed. For most purposes, this is not necessary. Here, for example, you could just make a simple and small table depicting the relevant results of the correlation analysis. |
Thank you for your comment. Corrected. |
|
19 |
The results section itself – this is a huge concern – is insufficient and does not correspond to the methods section. In the methods section you should say that you want to do t-tests, ANOVAs and chi-square tests. Where do we find them in the results section? In the methods section, you should say that in order to answer the research question, we want to do A, B, and C. Then in the results section, you should report the findings of A, B, and C. This is not what we see here. At the end, we even find a correlation matrix, which was never introduced in the methods section. So there are many questions emerging, like: is this a Pearson matrix? Or did they use Kendall’s Tau or Spearman’s Rho? |
Thank you for your comment. We have corrected methods and result sections according to your suggestions.
|
|
20 |
The discussion section is insufficient as well. You first need to quickly summarize what you did and what you wanted to find out. Then you should discuss your own findings and explain how they answered (or did not answer) your research questions. Consequently, you should then discuss how your findings correspond to the literature. Finally, some practical consequences can be mentioned. |
We have expanded the discussion according to your recommendations and inserted practical implications. |
|
21 |
I suggest inserting a conclusion chapter at the end and also moving the limitations in this section. |
Corrected. |
|
22 |
The perhaps biggest methodological shortcoming of this study is philosophical in nature. It has to do with the question what it is all about. The authors looked at variables concerning covid-19 stress and fear, eventually asking if online education is affected by this as well. This was predominantly analyzed via asking the respondents about their contentment with online education. However, looking at the effectiveness of online teaching is not necessarily measured best by simply asking students whether they are happy with the teaching mode. This is only one possible (albeit limited) approach to the question. The authors need to put in a lot more work convincing the readers that what they do actually corresponds to the validity of the study’s objectives. |
We especially thank you for the last two comments, which helped us a lot to restructure all sections, from the abstract to the conclusion. We hope that the clarifications we have made justify the set objectives of the study. |
|
23 |
Overall, this paper has major problems that are both methodological and redactional in nature. They are rooted very deep in the article. However, there is a lot of valuable data that the authors have worked on, and I believe that if the paper is thoroughly worked through once again and if considerable major revisions are applied (restructuring, redaction, language editing, and methodological strengthening of the arguments), the paper could be reconsidered for publication. |
Round 2
Reviewer 2 Report
The authors significantly improved the work. The work is almost completely redone. All praise for the effort invested. I suggest that the paper be accepted.
Reviewer 3 Report
The authors have made all the necessary changes to tge manuscript so that it would be ready for publication. Thank you for your work.